# Associations of SARS-CoV-2 antibodies with birth outcomes: Results from three urban birth cohorts in the NIH environmental influences on child health outcomes program

Leonardo Trasande[1,2,3,4,5]*, Sarah S. Comstock[6], Julie B. Herbstman[7], Amy Margolis[8], Garry Alcedo[1], Yelena Afanasyeva[1,2], Keunhyung Yu[1], William Lee[9,10], David A. Lawrence[9,10], on behalf of program collaborators for Environmental influences on Child Health Outcomes[¶]

1 Division of Environmental Pediatrics, Department of Pediatrics, NYU Grossman School of Medicine, New York, NY, United States of America, 2 Department of Population Health, NYU Grossman School of Medicine, New York, NY, United States of America, 3 Department of Environmental Health, NYU Grossman School of Medicine, New York, NY, United States of America, 4 NYU Wagner School of Public Service, New York, NY, United States of America, 5 NYU School of Global Public Health, New York, NY, United States of America, 6 Department of Food Science and Human Nutrition, Michigan State University, East Lansing, MI, United States of America, 7 Department of Environmental Health Sciences, Columbia Mailman School of Public Health, New York, NY, United States of America, 8 Department of Psychiatry, Columbia University Medical Center, New York, NY, United States of America, 9 Wadsworth Laboratories, New York State Department of Health, Albany, NY, United States of America, 10 School of Public Health, University at Albany, Albany, NY, United States of America

¶ See Acknowledgments for full listing of Environmental influences on Child Health Outcomes collaborators.
* leonardo.trasande@nyulangone.org

**Data Availability Statement:** Select de-identified data from the ECHO Program are available through NICHD's Data and Specimen Hub (DASH).

## Abstract

Studies suggest perinatal infection with SARS-CoV-2 can induce adverse birth outcomes, but studies published to date have substantial limitations. We therefore conducted an observational study of 211 births occurring between January 2020-September 2021 in three urban cohorts participating in the Environmental Influences on Child Health Outcomes Program. Serology was assessed for IgG, IgM and IgA antibodies to nucleocapsid, S1 spike, S2 spike, and receptor-binding domain. There were no differences in gestational age (GA), birth weight, preterm birth (PTB) or low birth weight (LBW) among seropositive mothers. However, the few (n = 9) IgM seropositive mothers had children with lower BW (434g, 95% CI: 116–752), BW Z score-for-GA (0.73 SD, 95% CI 0.10–1.36) and were more likely to deliver preterm (OR 8.75, 95% CI 1.22–62.4). Though there are limits to interpretation, the data support efforts to prevent SARS-CoV-2 infections in pregnancy.

## Background

The severe acute respiratory syndrome coronavirus 2 (SARS-CoV-2) pandemic continues with community spread throughout the United States, with over 80 million reported infections and nearly a million deaths [1]. SARS-CoV-2 is known to enter cells by binding the angiotensin-

Information on study data not available on DASH, such as some Indigenous datasets, can be found on the ECHO study DASH webpage. Data related queries may also be directed to the ECHO Data Analysis Center at ECHO-DAC@rti.org.

**Funding:** Research reported in this publication was supported by the Environmental influences on Child Health Outcomes (ECHO) program, Office of the Director, National Institutes of Health, under Award Numbers U2COD023375 (Coordinating Center), U24OD023382 (Data Analysis Center), U24OD023319 (PRO Core), UH3OD023282 (Gern), UH3OD023285 (Kerver), UH3OD023290 (Herbstman), UH3OD023305 (Trasande). The funders had no role in study design, data collection and analysis, decision to publish, or preparation of the manuscript.

**Competing interests:** The authors have declared that no competing interests exist.

**Abbreviations:** ACE2, Angiotensin-converting enzyme 2; CCCEH, Columbia Center for Children's Environmental Health; DSM-5, Diagnostic and Statistical Manual of Mental Disorders, 5th Edition; DBS, dried blood spots; ECHO, Environmental Influences on Child Health Outcomes Program; Ig, immunoglobulin; LBW, low-birth weight; N, nucleocapsid; CHES, NYU Children's Health and Environment Study; NYUGSOM, NYU Grossman School of Medicine; PTS, Pandemic-related Traumatic Stress; PTB, premature birth; RBD, receptor binding domain; SARS-CoV-2, severe acute respiratory syndrome coronavirus 2.

converting enzyme 2 (ACE2) receptor [2], which is heavily expressed in the placenta [3]. A large and diverse array of studies (including cohort, case-control and cross-sectional designs) has examined birth outcomes of babies born to mothers with SARS-CoV-2 infection or history of infection, identifying increases in preeclampsia, preterm birth and low birth weight [4, 5]. They are largely from the earliest (alpha) waves of the pandemic, and their exposure assessment is substantially heterogeneous, in that these meta-analyses included studies that leverage universal screening before or during labor and delivery, while others identify clinical cases based upon their presentation for clinical care. This heterogeneity may bias associations with categorical outcomes (such as preterm birth, PTB) to the null [6]. Most recently, a large surveillance-based study of Canadian women did suggest milder infections increased the risk of PTB compared to uninfected mothers, from 6.8% to 9.3% [7].

Another source of ambiguity in the effects of SARS-CoV-2 infection on birth outcomes is the presence of substantial psychosocial stress, which has been described in pregnant women during the pandemic [8], whether due to fear of infection, job loss, economic stress, psychological or physical trauma, or other factors. Psychological stress during pregnancy is associated with preterm birth [9], yet few studies have nested measures of SARS-CoV-2 infection within population-based cohorts of pregnant women to evaluate the effects of subclinical infection, independent of social and environmental determinants. One study has leveraged geospatial level data to contextualize structural racism and pandemic-related stress as well as seropositivity to evaluate joint effects on birth outcomes [10]. Though individual-level factors also drive disparities in reproductive outcomes [11], few studies have been able to integrate the multiple and potentially interacting factors and their contribution to birth outcomes.

The NIH Environmental Influences on Child Health Outcomes Program (ECHO) is a large national cohort program built pre-pandemic to examine preventable and environmental origins of health and disease in youth [12]. Built from existing observational cohorts of mothers and children, and largely representative of the US population [13], we present analyses of biological specimens collected from three participating cohorts across different waves of the pandemic (between January 2020-September 2021). Our primary aim was to evaluate relationships of SARS-CoV-2 serology, self-reported infection and pandemic-related stress with birth outcomes in largely urban cohorts from three ECHO cohort centers.

## Methods

### Study population

The present study was nested within a subsample of mothers and newborns enrolled into three participating ECHO cohorts. First, the NYU Children's Health and Environment Study (CHES), is a cohort of at least 2,000 mother-infant pairs recruited from three NYU Grossman School of Medicine (NYUGSOM) affiliate hospitals since 2016 at <18 weeks gestation [14]. Second, the Columbia Center for Children's Environmental Health (CCCEH) cohort, enrolled pregnant women from Columbia's New York Presbyterian Hospital ambulatory care network obstetric clinics beginning in 2013. Finally, the Michigan Archive for Research in Children's Health (MARCH) is a stratified random sample of births recruited in first trimester of pregnancy and supported by ECHO since 2017 [15]. Participating cohorts in the ECHO Program have been approved by Western IRB. Each participating mother provided informed and written consent. All methods were carried out in accordance with relevant guidelines and regulations. Within each cohort, a convenience sample of maternal and/or cord serum or dried blood spots (DBS) from births occurring between January 2020-September 2021 was analyzed. The period of sample collection was influenced by hospital system and other COVID-related precautions. We categorized the samples into four groups corresponding to waves of the

pandemic: Feb 28, 2020-June 19, 2020 (first wave); June 20, 2020-Sep 27, 2020 (second wave); Sep 28, 2020-Jan 10, 2021 (third wave); and after Jan 11, 2021 (fourth wave). We also categorized samples as being collected before or after the availability of SARS-CoV-2 vaccines (December 11, 2020).

### Evaluation of SARS-CoV-2 infection

Serum samples were diluted 1:100 with Phosphate Buffered Saline (PBS) plus 1% BSA and 0.05% sodium azide, pH 7.4 (PBN). For DBS punches PBS was used, which contained 0.1% BSA (Sigma-Aldrich, catalog #A-4503) and EDTA-free protease inhibitors (1 tablet/10 ml, Roche Applied Science, catalog #04693159001 (PBS$^+$). The DBS punches were eluted with PBS$^+$ and assayed for human immunoglobulin (Ig) isotypes as described [16]. Briefly, a 3.2 mm DBS punch was eluted in 60 μl of PBS$^+$ per well of a round bottom low-protein binding 96 well plate (catalog #1830–9600, USA Scientific) on an orbital shaker (Titer Plate Shaker, model # 1830–9600, USA Scientific) at 500 ± 50 rpm overnight at 4˚C. Elutes were combined from multiple wells, centrifuged 500 x g and supernatants used in the following assays. Diluted sera or eluted supernatants were split for immunoglobulin (Ig) isotype (IgG1-4, IgA, and IgM) quantification by Luminex200 with a HGAMMAG-301K Milliplex® kit (MilliporeSigma). For Ig isotypes to SARS-CoV antigens, a microsphere immunofluorescence assay (MIFA) like that described previously [17] was used. The MIFA has received US FDA EUA clearance. Specimens were assessed for the presence of antibody using the SARS-CoV-2 MIFA with carboxylated microspheres (Luminex Corp) coated with nucleocapsid (N), S1 spike (S), S2 S, and receptor-binding domain (RBD) antigen. Coupling was done per manufacturer's instructions using a purchased kit (Luminex Corp). Ig isotype antibodies were assayed with phycoerythrin (PE)-labelled goat antibodies to each isotype (Southern Biotech). Analysis was performed using a FlexMap 3D Analyzer (Luminex Corp).

### Assessment of infection

We first categorized mothers into three groups based upon their serology status (positive, indeterminate and negative). Sera collected from pre-SARS-CoV-2 (before 2019) healthy donors (n = 234) were used to calculate median values plus 3 standard deviation (SD) and 6 SD values for each immunoglobulin (Ig) isotype to each SARS-CoV-2 antigen. Samples collected for this study with values less than 3 SD from median were considered negative, those between the 3 SD and 6 SD values were classified as indeterminant, and those above the 6 SD value were consider positive. Positive mothers had positivity to any of the immunoglobulins (IgG1-4, IgM, IgA) to N, S1, S2, or RBD. Indeterminate mothers did not have any value for immunoglobulins above 6 SD but at least one indeterminate result (3–6 SD). Negative mothers had no indeterminate or positive result for any of the antigens.

We also created separate variables to classify mothers by their IgM serology results as indeterminate, positive or negative. A similar approach was used to categorize mothers for IgG (consolidating IgG1-4 results) and IgA, respectively. Finally, we created separate variables to classify antibody response as indeterminate, positive or negative separately for each of the four antigens (N, S1, S2 and RBD, respectively).

The ECHO Program also implemented a COVID-19 questionnaire, asking mothers to report infection. Primary analyses integrated these results with self-report of infection (either by test or diagnosis). Mothers with self-report of infection were categorized as positive in main analyses regardless of their serology.

## Assessment of pandemic-related traumatic stress

Investigators from the ECHO program also developed the PTS Scale based on the Diagnostic and Statistical Manual of Mental Disorders, 5th Edition (DSM-5) Acute Stress Disorder criteria [18]. The nine-item scale was developed with the express purpose of measuring an individual's experience of acute stress symptoms induced by the COVID-19 pandemic and associated social isolation related to mitigation actions. These items evaluate symptomology and severity within the month following a traumatic event. Although living during a pandemic may not be traditionally viewed as an inciting event in the DSM-5 (American Psychiatric Association, 2013), the COVID-19 pandemic did present as a life-threatening experience–both real and perceived–to many individuals. Because the items were completed more than one month after the onset of the pandemic, the responses are thought to better reflect pandemic-related traumatic stress (PTS) than acute stress.

Items were developed based on the DSM-5 acute stress criteria [19], and the scale was designed to map onto each of the five DSM-5 Acute Stress Disorder symptom categories: (1) intrusion (e.g., distressing memories and dreams, flashbacks, catastrophized perceptions of expected events or conditions); (2) negative mood (e.g., anhedonia, anger disproportionate to the situation); (3) dissociation (e.g., feelings of time slowing); (4) avoidance (e.g., purposeful efforts to avoid thinking about the event or actions that are not congruent with required realities of persisting threats); and (5) difficulty regulating arousal (e.g., sleep disturbance, irritability, poor concentration). These concepts were written into survey item format based on examples given in the DSM-5 text, and items were discussed by the ECHO research team to reach consensus on the clarity of the items. Mokken analysis resulted in a moderately strong, unidimensional scale based upon nine items ($\alpha > 0.8$). Scores were summed across the nine items, and the PTS Scale was considered as a continuous variable.

## Outcomes and covariates

We evaluated gestational age (GA), birth weight (BW) and BW Z-score for gestational age (using INTERGROWTH criteria [20]) as continuous variables. Categorical outcomes included prematurity ($<37$ weeks), low birth weight (LBW, $<2500$ g) and small for gestational age (SGA). We considered covariates including maternal age, parity, sex of child and maternal education as measured using ECHO-wide data collection forms. Due to the limited sample size and potential overspecification of models, we did not include medical conditions which can affect birthweight, such as hypertension, diabetes, obesity and rheumatologic diseases. We also did not add time to infection as a covariate to limit overspecification of models.

## Statistical analyses

After describing the sample, we performed Pearson correlation coefficients of infection, serology (categorized as positive, indeterminate or negative) and stress. We then performed univariate and multivariable analyses of GA, BW and BW Z-score for gestational age, as well as PTB, SGA and LBW, comparing uninfected, indeterminate and infected mothers. We repeated the analyses substituting serology status (across all antigens and immunoglobulin subtypes, and then by antigen and immunoglobulin subtype). We then performed univariate and multivariable analyses of birth outcomes against PTS symptoms in univariate as well as multivariable models sequentially adding infection and then sociodemographic variables. Stratification by sex and stress was not performed due to overspecification of some models. Univariate regressions were repeated within the subset collected before December 11, 2020 (pre-vaccine) as well as in subsets representing the first and fourth waves.

## Results

The study population (Table 1) was nearly equally distributed between New York and Michigan, with 36% positivity by serology and 43% with infection identified by self-report or serology. IgG positivity was the chief driver of seropositivity (34%). Eleven participant mothers were seronegative but reported infection, as did two mothers with at least one indeterminate antibody test (S1 Table in S1 File). The income and education distributions were generally bimodal, and the population was 23% Hispanic and 15% non-Hispanic Black. The mean maternal age was 30.7 (SD 5.7) years and the sample was relatively healthy in terms of birth outcomes, with lower percentages of prematurity, SGA and LBW compared to national rates.

Self-report of SARS-CoV-2 infection was not significantly correlated with combined serology (S2 Table in S1 File) but was modestly correlated with IgG1 serology (categorized as positive/indeterminate/negative; r = 0.17, p = 0.013) as well as N (r = 0.16, p = 0.024) and RBD (r = 0.14, p = 0.046) serology. Correlations between serotypes and antigen specificity were moderate-to-high. PTS symptoms were not correlated with the composite self-reported or serologically confirmed evidence of prior exposure to or infection with SARS-CoV-2.

In regression models comparing infected to uninfected mothers as defined with the composite measure inclusive of serology and self-report (Table 2A), there were no differences in gestational age, birth weight, PTB or LBW. Nearly identical results were obtained when comparing birth outcomes by serology status (Table 2B). IgM seropositive mothers in the sample (n = 9, Table 2C) had children with 407 grams lower birth weight (95% CI: 85–728) and BW Z-score for GA (0.65 SD; 95% CI 0.002–1.30). They were also more likely to deliver preterm (OR 6.29, 95% CI 1.12–25.2) in univariate models. Addition of covariates did not change effect size for IgM seropositivity (434g lower, 95% CI: 116–752 for birth weight; 0.73 SD lower birth weight Z-score for gestational age, 95% CI 0.10–1.36; OR 8.75, 95% CI 1.22–62.4 for PTB).

Indeterminate IgM status was not associated with significant differences in birth outcomes. There was also an increase in PTB noted among children born to mothers with indeterminate IgG status (S3 Table in S1 File; OR 7.76, 95% CI: 1.38–51.6 in multivariable models), but IgG seropositive mothers did not give birth to children with significantly different birth outcomes. Analyses by antigen response (S4 Table in S1 File) failed to reveal any differences in birth outcomes.

PTS symptom data were available in over 60% of the sample. Within this subsample, the composite of seropositivity or self-report infection was not associated with birth outcomes (Table 3). When PTS symptoms was added to regressions of birth outcomes in IgM seropositive mothers to other mothers, decreases in birth weight (-426 g, 95% CI -820, -31) sustained, while PTB was not significantly albeit tended to be positively associated (OR 16.2, 95% CI 0.58, 447).

Sensitivity analyses of the subset of pre-vaccine samples (Table 4A-4C) failed to reveal differences by composite infection or serology in any birth outcomes. IgM positivity among pre-vaccine samples was significantly associated with low birth weight, with substantial negative associations (nonsignificant) for continuous birth weight and gestational length and suggestive increases in prematurity. Similarly, analyses within the first and fourth waves failed to reveal any associations of composite infection or serology in any birth outcomes (Table 4D-4I). In the first wave, IgM positivity was associated with significantly increased preterm birth and suggestive decreases in gestational age and birth weight as well as increases in low birth weight. In the fourth wave, IgM positivity was not significantly associated with any outcomes, although suggestive increases in low birth weight and decreases in continuous birth weight were noted.

**Table 1. Description of study population (N = 211).**

| **Study Site** | | | |
|---|---|---|---|
| | NYU Children's Health and Environment Study | 89 | 42% |
| | Michigan Archive for Research on Child Health | 103 | 49% |
| | Columbia Children's Environmental Health and Environment Studies | 19 | 9% |
| Serology Results | | | |
| | IgM/G/A positive to at least one antigen, N (%) | 77 | 36% |
| | Indeterminate to at least one antigen, N (%) | 29 | 14% |
| | Negative serology to all antigens, N (%) | 105 | 50% |
| | IgM positive to at least one antigen, N (%) | 9 | 4% |
| | IgA positive to at least one antigen, N (%) | 41 | 19% |
| | IgG positive to at least one antigen, N (%) | 71 | 34% |
| | Any antibody response to N antigen, N (%) | 21 | 10% |
| | Any antibody response to S1 antigen, N (%) | 58 | 27% |
| | Any antibody response to S2 antigen, N (%) | 55 | 26% |
| | Any antibody response to RBD antigen, N (%) | 56 | 27% |
| Positivity to at least one antigen, N (%) | | | |
| | Feb 28, 2020-June 19, 2020 (first wave, N = 83) | 17 | 20% |
| | June 20, 2020-Sep 27, 2020 (second wave, N = 20) | 7 | 35% |
| | Sep 28, 2020-Jan 10, 2021 (third wave, N = 24) | 7 | 29% |
| | Jan 11, 2021-May 31, 2021 (fourth wave, N = 84) | 46 | 55% |
| | Pre-vaccine (before December 11, 2020, N = 116) | 27 | 23% |
| Positive serology or self-report infection, N (%) | | 90 | 43% |
| Household Income (N = 194) | | | |
| | Less than $49,999 | 68 | 35% |
| | $50,000-$74,999 | 11 | 6% |
| | $75,000-$99,999 | 16 | 8% |
| | $100,000 or more | 89 | 46% |
| | Don't know | 10 | 5% |
| Maternal education (N = 198) | | | |
| | High school or less | 49 | 25% |
| | Some college but no degree | 29 | 15% |
| | Associate degree | 11 | 6% |
| | Bachelor's degree | 55 | 28% |
| | Post-graduate degree | 54 | 27% |
| Race/ethnicity | | | |
| | Hispanic | 49 | 23% |
| | Non-Hispanic White | 108 | 51% |
| | Non-Hispanic Black | 31 | 15% |
| | Asian | 10 | 5% |
| | Other | 6 | 3% |
| | Multiple race | 7 | 3% |
| Parity | | | |
| | 0 | 97 | 46% |
| | ≥1 | 114 | 54% |
| Child Sex | | | |
| | Male | 103 | 49% |
| | Female | 108 | 51% |
| Maternal employment | | | |

*(Continued)*

**Table 1.** (Continued)

| Study Site | | | | |
|---|---|---|---|---|
| | Not employed | | 47 | 22% |
| | Employed | | 164 | 78% |
| Stress Scale (N = 122), Median (IQR) | | | 19 | (14, 24.25) |
| Birth Weight, Mean (SD) | | | 3294 | (482) |
| Gestational Age, Mean (SD) | | | 38.9 | (1.57) |
| Prematurity, % | | | | 5.2% |
| Low Birth Weight, % | | | | 3.8% |
| Maternal Age, Mean (SD) | | | 30.7 | (5.7) |
| Weight for Gestational Age Z-Score, Median (IQR) | | | 0.41 | (-0.29, 1.05) |
| Small for Gestational Age, % | | | 12 | 5.7% |

## Discussion

The present study associated IgM seropositivity to SARS-CoV-2 with decreased birth weight, BW for GA, and increased prematurity in largely urban cohorts from three cohort centers participating in ECHO. PTS symptoms were not associated with birth outcomes, nor were composite assessments of exposure integrating self-report of infection. The effect size of IgM seropositivity with adverse birth outcomes was substantial including in the pre-vaccine

**Table 2. a.** Comparing Birth Outcomes in Mothers by Composite Serology/Self-Report (N = 211). **b.** Comparing Birth Outcomes in Mothers by Composite Serology (N = 211). **c.** Comparing Birth Outcomes in Mothers by IgM Subtype (N = 211).

| | Univariate | | Multivariable | |
|---|---|---|---|---|
| | Indeterminate | Seropositive/Self-Report Infected | Indeterminate | Seropositive/Self-Report Infected |
| **a** | | | | |
| Gestational Age, weeks (95% CI) | 0.08 (-0.60, 0.76) | -0.06 (-0.52, 0.40) | 0.03 (-0.66, 0.72) | -0.09 (-0.55, 0.37) |
| Birth Weight, grams (95% CI) | -65 (-275, 144) | -34 (-174, 107) | -27 (-236, 182) | -51 (-192, 89) |
| Preterm Birth, OR (95% CI) | 2.66 (0.56, 12.7) | 1.04 (0.25, 4.32) | 2.79 (0.51, 15.2) | 1.03 (0.23, 4.67) |
| Low Birth Weight, OR (95% CI) | 1.10 (0.11, 11.1) | 0.93 (0.23, 3.85) | 0.92 (0.09, 9.78) | 1.23 (0.26, 5.92) |
| Weight for Gestational Age Z-score (95% CI) | -0.16 (-0.57, 0.24) | -0.01 (-0.30, 0.27) | -0.04 (-0.44, 0.35) | 0.001 (-0.28, 0.29) |
| Small for Gestational Age, OR (95% CI) | 0.43 (0.05, 3.61) | 0.49 (0.13, 1.92) | 0.38 (0.04, 3.42) | 0.50 (0.11, 2.18) |
| **b** | | | | |
| Gestational Age, weeks (95% CI) | 0.05 (-0.60, 0.70) | -0.10 (-0.56, 0.37) | -0.01 (-0.68, 0.65) | -0.19 (-0.67, 0.29) |
| Birth Weight, grams (95% CI) | -77 (-278, 123) | -21 (-164, 122) | -38 (-238, 160) | -27 (-170, 117) |
| Preterm Birth, OR (95% CI) | 2.91 (0.61, 13.8) | 1.38 (0.34, 5.71) | 2.90 (0.55, 15.4) | 1.35 (0.30, 6.08) |
| Low Birth Weight, OR (95% CI) | 1.21 (0.12, 12.2) | 1.86 (0.40, 8.58) | 1.02 (0.10, 10.7) | 1.71 (0.36, 8.25) |
| Weight for Gestational Age Z-score (95% CI) | -0.15 (-0.57, 0.27) | -0.10 (-0.37, 0.18) | -0.03 (-0.44, 0.38) | -0.10 (-0.37, 0.18) |
| Small for Gestational Age, OR (95% CI) | + | 0.93 (0.29, 3.00) | + | 1.02 (0.29, 3.70) |
| **c** | | | | |
| Gestational Age, weeks (95% CI) | -0.05 (-0.81, 0.72) | -0.64 (-1.70, 0.41) | 0.08 (-0.87, 0.71) | -0.76 (-1.83, 0.31) |
| Birth Weight, grams (95% CI) | 7.9 (-225, 240) | -407 (-728, -85)* | 66 (-168, 300) | -434 (-752, -116)** |
| Preterm Birth, OR (95% CI) | 1.30 (0.15, 11.0) | 6.29 (1.12, 35.2)* | 1.19 (0.12, 11.3) | 8.75 (1.22, 62.4)* |
| Low Birth Weight, OR (95% CI) | 1.75 (0.20, 15.4) | 3.71 (0.40, 34.6) | 1.51 (0.16, 14.4) | 4.15 (0.40, 43.2) |
| Weight for Gestational Age Z-score (95% CI) | 0.08 (-0.38, 0.55) | -0.65 (-1.30, -0.002)* | 0.13 (-0.33, 0.60) | -0.73 (-1.36, -0.10)* |
| Small for Gestational Age, OR (95% CI) | 2.18 (0.44, 10.8) | + | 3.78 (0.61, 23.4) | + |

Multivariable models control for maternal age, parity, sex of child and maternal education.

+ All mothers gave birth to non-SGA newborns.

**Table 3. Associations of pandemic stress with birth outcomes (N = 122).**

|  | Univariate | Controlled for Infection Status | Multivariable |
|---|---|---|---|
| Gestational Age, weeks (95% CI) | 0.03 (-0.01, 0.06) | 0.03 (-0.01, 0.06) | 0.04 (-0.002, 0.07) |
| Birth Weight, grams (95% CI) | -3 (-16, 9) | -4 (-16, 9) | -2 (-15, 11) |
| Preterm Birth, OR (95% CI) | 0.85 (0.70, 1.04) | 0.86 (0.70, 1.04) | 0.81 (0.63, 1.05) |
| Low Birth Weight, OR (95% CI) | 1.14 (0.96, 1.33) | 1.16 (0.97, 1.38) | 1.65 (0.12, 22.8) |
| Weight for Gestational Age Z-score (95% CI) | -0.02 (-0.05, 0.01) | -0.02 (-0.05, 0.01) | -0.02 (-0.05, 0.01) |
| Small for Gestational Age, OR (95% CI) | 1.07 (0.96, 1.20) | 1.07 (0.96, 1.20) | 1.10 (0.97, 1.26) |

Multivariable models also control for maternal age, parity, sex of child and maternal education

subsample, while overall seropositivity was not associated with birth outcomes in the entire sample or in pre-vaccine or wave-specific subsets.

As for most immune responses including those to a viral infection such as SARS-CoV-2, innate immunity precedes adaptive immunity and antibody production. Once B cell responses begin IgM usually precedes IgG production unless prior memory existed. IgM levels during COVID-19 have been reported to be non-detectable during the first 10 days following infection, with more severe patients observed to have higher levels of IgM [21]. In a different study based on the ELISA [22] and MIFA [23] methods we used to assess serology to SARS-CoV-2, the IgM and IgG levels were detectable within 10 days, but IgM and IgG appeared simultaneously, and IgG levels lasted longer. IgM and IgG to N and S protein of SARS-CoV-2 peaked between 3–5 weeks after infection but by 12 weeks the IgM levels declined to a near negative level whereas IgG remained detectable [24]. The associations with IgM seropositivity for lower BW and greater PTB observed here should be interpreted in the context of the small sample size (N = 9) and the absence of a matching relationship with anti-N antibody, which has also been described as a signature of infection rather than vaccination [25].

The findings should be interpreted with modesty but are supportive of other associations of infection with adverse birth outcomes identified from clinically-recruited populations [4, 5] as well as population-based studies [7]. The data also support ongoing efforts to prevent SARS-CoV-2 infections in pregnancy. Looking beyond fetal growth and gestational length, the potential exists for other long-term consequences of prenatal SARS-CoV-2 infection. Cohorts participating in the ECHO program have archived samples and collected other relevant covariates and confounders before and during the pandemic. They are well poised to document the presence or absence of disease and disability of early life exposure to SARS-CoV-2 infection. With a harmonized protocol and a sample broadly representative of the United States, findings from ECHO and other large populations will be crucial to shaping long-term care of children with antecedent SARS-CoV-2 infection during gestation.

Strengths of the study include the joint examination of PTS symptoms and infection, leverage of existing diverse and well-characterized population-based cohorts from more than one urban center and use of a rapidly implemented COVID-19 questionnaire to document effects of the pandemic, including infection. As a result, the present manuscript includes data from multiple waves of the pandemic, and we were able to use multiple methods including serologic testing to ascertain infection, and effects independent of harmonized covariates and potential confounders. The modest correlation of infection ascertained by self-report with antibody testing supports the use of biological specimens to ascertain infection and timing in further studies of prenatal infection with SARS-CoV-2.

We emphasize that the relatively small sample size produced associations with wide confidence intervals and further limited our ability to examine potential sex-stratified and other

**Table 4.** **a**. Comparing Birth Outcomes in Mothers by Composite Serology/Self-Report Before Vaccine EUA (N = 116). **b**. Comparing Birth Outcomes in Mothers by Composite Serology Before Vaccine EUA (N = 116). **c**. Comparing Birth Outcomes in Mothers by IgM Serotype Before Vaccine EUA (N = 116). **d**. Comparing Birth Outcomes in Mothers by Composite Serology/Self-Report in First Wave (N = 83). **e**. Comparing Birth Outcomes in Mothers by Composite Serology/Self-Report in Fourth Wave (N = 84). **f**. Comparing Birth Outcomes in Mothers by Composite Serology in First Wave (N = 83). **g**. Comparing Birth Outcomes in Mothers by Composite Serology in Fourth Wave (N = 84). **h**. Comparing Birth Outcomes in Mothers by IgM Serology in First Wave (N = 83). **i**. Comparing Birth Outcomes in Mothers by IgM Serology in Fourth Wave (N = 84).

| | Univariate | |
|---|---|---|
| **a** | | |
| | Indeterminate | Seropositive/Self-Report Infected |
| Gestational Age, weeks (95% CI) | 0.21 (-0.54, 0.97) | 0.16 (-0.50, 0.82) |
| Birth Weight, grams (95% CI) | 26 (-218, 271) | -69 (-283, 144) |
| Preterm Birth, OR (95% CI) | 2.37 (0.37, 15.3) | 1.58 (0.25, 10.0) |
| Low Birth Weight, OR (95% CI) | 1.71 (0.15, 19.9) | 3.75 (0.59, 23.8) |
| Weight for Gestational Age Z-score (95% CI) | -0.01 (-0.49, 0.46) | -0.15 (-0.58, 0.28) |
| Small for Gestational Age, OR (95% CI) | 0.63 (0.07, 5.71) | 1.57 (0.35, 7.11) |
| **b** | | |
| | Indeterminate | Seropositive |
| Gestational Age, weeks (95% CI) | 0.22 (-0.51, 0.97) | 0.17 (-0.50, 0.84) |
| Birth Weight, grams (95% CI) | 6 (-234, 247) | -32 (-251, 187) |
| Preterm Birth, OR (95% CI) | 1.65 (0.14, 19.2) | 4.13 (0.65, 26.2) |
| Low Birth Weight, OR (95% CI) | 2.28 (0.35, 14,7) | 1.73 (0.27, 11.0) |
| Weight for Gestational Age Z-score (95% CI) | 0.05 (-0.43, 0.53) | -0.25 (-0.67, 0.17) |
| Small for Gestational Age, OR (95% CI) | + | 1.98 (0.49, 8.00) |
| **c** | | |
| | Indeterminate | Seropositive |
| Gestational Age, weeks (95% CI) | 0.48 (-0.48, 1.44) | -1.86 (-3.94, 0.22) |
| Birth Weight, grams (95% CI) | -22 (-335, 291) | -684 (-1360, 8) |
| Preterm Birth, OR (95% CI) | + | 16.3 (0.90, 294) |
| Low Birth Weight, OR (95% CI) | 2.78 (0.28, 28) | 25.0 (1.31, 476) |
| Weight for Gestational Age Z-score (95% CI) | -0.16 (-0.78, 0.46) | -0.97 (-2.31, 0.37) |
| Small for Gestational Age, OR (95% CI) | 3.46 (0.61, 19.5) | + |
| **d** | | |
| | Indeterminate | Seropositive |
| Gestational Age, weeks (95% CI) | -0.35 (-1.36, 0.65) | -0.09 (-1.36, 0.65) |
| Birth Weight, grams (95% CI) | -92 (-429, 243) | -133 (-405, 137) |
| Preterm Birth, OR (95% CI) | 5.20 (0.30, 90.2) | 6.12 (0.52, 71.8) |
| Low Birth Weight, OR (95% CI) | 2.55 (0.21, 30.9) | 4.78 (0.73, 31.2) |
| Weight for Gestational Age Z-score (95% CI) | 0.01 (-0.62, 0.65) | -0.31 (-0.82, 0.20) |
| Small for Gestational Age, OR (95% CI) | + | 2.30 (0.46, 11.4) |
| **e** | | |
| | Indeterminate | Seropositive |
| Gestational Age, weeks (95% CI) | -0.82 (-2.71, 1.06) | -0.05 (-0.86, 0.97) |
| Birth Weight, grams (95% CI) | -440 (-986, 106) | -2 (-266, 262) |
| Preterm Birth, OR (95% CI) | + | 0.28 (0.02, 4.69) |
| Low Birth Weight, OR (95% CI) | 5.67 (0.27, 117) | 0.57 (0.05, 6.63) |
| Weight for Gestational Age Z-score (95% CI) | -0.88 (-1.98, 0.22) | -0.14 (-0.67, 0.39) |
| Small for Gestational Age, OR (95% CI) | + | + |
| **f** | | |
| | Indeterminate | Seropositive |

(*Continued*)

**Table 4.** (Continued)

| | Univariate | |
|---|---|---|
| Gestational Age, weeks (95% CI) | 0.28 (-1.26, 0.68) | -0.10 (-0.94, 0.74) |
| Birth Weight, grams (95% CI) | -121 (-445, 203) | -85 (-366, 198) |
| Preterm Birth, OR (95% CI) | 2.36 (0.20, 28.4) | 5.57 (0.85, 36.7) |
| Low Birth Weight, OR (95% CI) | 4.82 (0.28, 83.0) | 7.07 (0.60, 83.3) |
| Weight for Gestational Age Z-score (95% CI) | -0.11 (-0.72, 0.51) | -0.16 (-0.70, 0.37) |
| Small for Gestational Age, OR (95% CI) | 1.14 (0.12, 11.2) | 1.67 (0.28, 10.0) |
| **g** | | |
| | Indeterminate | Seropositive |
| Gestational Age, weeks (95% CI) | -0.99 (-2.62, 0.63) | -0.24 (-1.01, 0.54) |
| Birth Weight, grams (95% CI) | -371 (-846, 104) | -3 (-229, 223) |
| Preterm Birth, OR (95% CI) | + | 0.71 (0.04, 11.8) |
| Low Birth Weight, OR (95% CI) | 8.00 (0.41, 154) | 1.45 (0.13, 16.7) |
| Weight for Gestational Age Z-score (95% CI) | -0.58 (-1.54, 0.38) | 0.07 (-0.39, 0.53) |
| Small for Gestational Age, OR (95% CI) | + | + |
| **h** | | |
| | Indeterminate | Seropositive |
| Gestational Age, weeks (95% CI) | 0.27 (-1.11, 1.65) | -1.80 (-3.94, 0.34) |
| Birth Weight, grams (95% CI) | -104 (-565, 357) | -662 (-1377, 53) |
| Preterm Birth, OR (95% CI) | + | 24.3 (1.21, 490) |
| Low Birth Weight, OR (95% CI) | 4.50 (0.40, 50.2) | 18.0 (0.94, 343) |
| Weight for Gestational Age Z-score (95% CI) | -0.23 (-1.11, 0.65) | -0.94 (-2.31, 0.43) |
| Small for Gestational Age, OR (95% CI) | 2.92 (0.28, 30.4) | + |
| **i** | | |
| | Indeterminate | Seropositive |
| Gestational Age, weeks (95% CI) | -1.47 (-3.02, 0.08) | -0.17 (-1.60, 1.25) |
| Birth Weight, grams (95% CI) | -87 (-370, 543) | -340 (-760, 78) |
| Preterm Birth, OR (95% CI) | + | + |
| Low Birth Weight, OR (95% CI) | 8.88 (0.65, 120) | 7.1 (0.54, 92.4) |
| Weight for Gestational Age Z-score (95% CI) | 0.68 (-0.23, 1.59) | -0.63 (-1.46, 0.20) |
| Small for Gestational Age, OR (95% CI) | + | + |

interaction effects on fetal growth. We were also unable to evaluate potential windows of susceptibility. Specifically, we could not evaluate infection prior to conception, as our study design did not permit evaluation of pregnancy loss. The generalizability is also limited to urban subpopulations. The majority of the sample was evaluated with serology before the vaccine EUA on December 11, 2020, but we cannot rule out residual confounding by vaccination status. This study was unable to directly confirm whether seropositivity was the byproduct of vaccine or infection, or both. Stratification of the sample to focus on the pre-vaccine period mostly corroborated the main findings. Our evaluation of PTS symptomatology together with infection was only possible in 122 study participants, and we make the distinction here of PTS with other sources of stress that may modify the effects of infection. Finally, we were unable to include in models conditions which confound effects on birth weight, and time to vaccination, as covariates due to model overspecification.

## Conclusions

We identified decreased birth weight, BW Z-score for GA and increased prematurity in mothers IgM seropositive to SARS-CoV-2, independent of PTS. Though the sample size is limited, the data support efforts to prevent SARS-CoV-2 infections in pregnancy, and future studies of later-life effects.

## Supporting information

**S1 File. Cross tabulations of serology against self-report of infection; Pearson correlations of stress and infection and serology in the study population; Comparing birth outcomes in mothers by serology subtype (N = 211); Comparing birth outcomes in mothers by antigen response (N = 211).**
(DOCX)

## Acknowledgments

The abstract of this manuscript in a preliminary form was presented at the Society for Birth Defects Research and Prevention on June 27, 2023 and is available online at https://onlinelibrary.wiley.com/doi/10.1002/bdr2.2030.

**ECHO collaborators acknowledgments**

The authors wish to thank our ECHO colleagues; the medical, nursing, and program staff; and the children and families participating in the ECHO cohorts. We also acknowledge the contribution of the following ECHO program collaborators:

ECHO Components—Coordinating Center: Duke Clinical Research Institute, Durham, North Carolina: Smith PB, Newby KL; Data Analysis Center: Johns Hopkins University Bloomberg School of Public Health, Baltimore, Maryland: Jacobson LP; Research Triangle Institute, Durham, North Carolina: Catellier DJ; Person-Reported Outcomes Core: Northwestern University, Evanston, Illinois: Gershon R, Cella D.

ECHO Awardees and Cohorts—University of Wisconsin, Madison WI: Gern J; Columbia University Medical Center, New York, NY: Miller R; Michigan State University, East Lansing, MI: Kerver J; Henry Ford Health System, Detroit, MI: Barone, C; Michigan Department of Health and Human Services, Lansing, MI: McKane, P; Michigan State University, East Lansing, MI: Paneth N; University of Michigan, Ann Arbor, MI: Elliott, M.

## Author Contributions

**Conceptualization:** Leonardo Trasande.

**Data curation:** Leonardo Trasande, Sarah S. Comstock, Julie B. Herbstman, Yelena Afanasyeva, Keunhyung Yu, William Lee, David A. Lawrence.

**Formal analysis:** Leonardo Trasande.

**Funding acquisition:** Leonardo Trasande.

**Investigation:** Leonardo Trasande, Sarah S. Comstock, Julie B. Herbstman, Amy Margolis.

**Methodology:** Garry Alcedo, William Lee, David A. Lawrence.

**Project administration:** Leonardo Trasande, Garry Alcedo.

**Writing – original draft:** Leonardo Trasande.

**Writing – review & editing:** Sarah S. Comstock, Julie B. Herbstman, Amy Margolis, Yelena Afanasyeva, Keunhyung Yu.

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
