## [Decision Letter · Decision Letter 0]

21 Sep 2023

PONE-D-23-20437Associations of SARS-CoV-2 Antibodies with Birth Outcomes: Results from Three Urban Birth Cohorts in the NIH Environmental Influences on Child Health Outcomes ProgramPLOS ONE

Dear Dr. Trasande,

Thank you for submitting your manuscript to PLOS ONE. After careful consideration, we feel that it has merit but does not fully meet PLOS ONE’s publication criteria as it currently stands. Therefore, we invite you to submit a revised version of the manuscript that addresses the points raised during the review process.

We look forward to receiving your revised manuscript.

Kind regards,

Faten Abdelaal Okda

Academic Editor

PLOS ONE

5. One of the noted authors is a group or consortium [program collaborators for Environmental influences on Child Health Outcomes]. In addition to naming the author group, please list the individual authors and affiliations within this group in the acknowledgments section of your manuscript. Please also indicate clearly a lead author for this group along with a contact email address.

7. Please upload a copy of Supporting Information (Supplementary Table 1,2,3,4) which you refer to in your text on page 11 and 12.

Reviewers' comments:

Reviewer's Responses to Questions

**Comments to the Author**

1. Is the manuscript technically sound, and do the data support the conclusions?

Reviewer #1: Yes

Reviewer #2: Yes

2. Has the statistical analysis been performed appropriately and rigorously? 

Reviewer #1: Yes

Reviewer #2: Yes

3. Have the authors made all data underlying the findings in their manuscript fully available?

Reviewer #1: Yes

Reviewer #2: Yes

4. Is the manuscript presented in an intelligible fashion and written in standard English?

Reviewer #1: Yes

Reviewer #2: Yes

5. Review Comments to the Author

Reviewer #1: The manuscript presents an attempt to evaluate the relationships of SARS-CoV-2 serology, self-reported infection, and pandemic-related stress with birth outcomes in largely urban cohorts from three ECHO cohort centers.

The pregnancy outcomes are well-defined and include gestational age (GA), birth weight (BW), and BW Z-score for gestational age (as continuous variables. Categorical outcomes included prematurity,

low birth weight (LBW) and small for gestational age (SGA).

The analyses did not document any differences in gestational age, birth weight, PTB, or LBW. Nearly identical results were obtained when comparing birth outcomes by serology status. The only positive association was the finding that children with lower birth weight and BW Z-score for GA (were more likely to deliver preterm in univariate models.

The study is performed in an elegant way and I do not have any major questions regarding the methodology.

The only point worth explaining is the way the final sample was identified. It is said that the study was nested within a subsample of mothers and newborns enrolled in three participating ECHO cohorts. Why three different cohorts were utilized? What were the criteria for inclusion and exclusion?

Reviewer #2: Here the authors aim at understanding the effect that SARS-CoV-2 infection has during or before pregnancy. In an observational study including three urban cohorts that gave birth between January 2020 to September 2021 they analyzed samples from 211 mothers for the presence of IgG1, IgG2, IgG3, IgG4, IgM and IgA subclass antibodies against N, S1, S2 and RBD areas of the spike protein. Pre-pandemic samples (n=234) were used to define non-infected antibody levels, and levels between 3-6 SD above background were classed as indeterminate whereas over 6 SD were considered positive. All mother with at least one subclass over 6SD were considered positive, mothers with a least on subclass in the 3-6 SD as indeterminate, and all classes below 3SD as negative. Seropositivity was seen in 36% of mothers, while 43% reported that they had been infected. IgG1 total seropositivity as well as reactivity against N and RBD was modestly correlated with reported infection. Neither serology + self reported infection nor serology alone was correlated with gestational age, birth weight, pre-term birth or low birth weight. IgM positivity was scored in 9 mothers, and in this case a significant decrease in birth weight was observed. This could not be solely explained by a posttraumatic stress score calculated from the mothers. In addition, in children born from mothers before any vaccines was available, decreased birth weights was also seen children born to IgM positive mothers.

During COVID-19 infection it has been shown that IgM antibodies occur in serum at approximately the same time as IgG and IgA antibodies. Since IgM then decrease more rapidly the data indicate that recent (possibly during pregnancy) infections with COVID-19 can influence pregnancy outcomes, but the authors argue caution as this result is only based on very few cases (n=9).

Overall, the study is of interest although its main conclusion is based on very few samples. It is well written, the data is presented in a clear and careful way, and the results are discussed cautiously. Although the data will not overall change the general recommendation to try to avoid infection with SARS-CoV- 2 during pregnancy, it gives some further support to the recommendation.

Specific points

I think the material is big enough to not only discuss the potential correlation between IgM antibodies and pregnancy/birth related problems but also the lack of any correlation to IgG levels. This could suggest that it is unlikely that pre-pregnancy (or even early pregnancy) infections will not interfere with outcomes. I fully understand that it is hard to rule out that this could be due to early problems are not scored (i.e. problems conceiving or early spontaneous abortion), but at least late problems from previous infection are unlikely. I would suggest a short paragraph about this in the discussion.

On line 234, the sentence “When we compared mothers by immunoglobulin subtype (Table 2c), the small number of IgM seropositive mothers in the sample (n=9) had children with 407 grams lower birth weight (95% CI: 85-728) and BW Z-score for GA (0.65 SD; 95% CI 0.002-1.30) were more likely to deliver preterm (OR 6.29, 95% CI 1.12-25.2) in univariate models.” Is very hard to understand – please reformulate.

6. PLOS authors have the option to publish the peer review history of their article (what does this mean?). If published, this will include your full peer review and any attached files.

Reviewer #1: **Yes: **Wojciech Hanke

Reviewer #2: No

---

## [Author Response · Author response to Decision Letter 0]

11 Oct 2023

Faten Abdelaal Okda

Academic Editor

PLOS ONE

Thanks for the thoughtful reviewer comments regarding our manuscript.

We address the journal requirements, editorial comments and reviews below point-by-point.

We look forward to further consideration and acceptance by PLOS ONE.

Sincerely,

Leonardo Trasande, MD, MPP on behalf of the coauthors

We have done as requested.

We have reviewed the manuscript, and these sections match. In addition, we have reviewed the FUnding Information in the PLOS ONE system to ensure alignment.

Sorry – there seems to be a misunderstanding. The NIH ECHO Program has policies about sharing the data through the NICHD Data and Specimen Hub. We cannot provide data directly. Because the NIH funded the research, we must follow the legal requirement of our funder.

The ORCID iD for the author has been updated.

5. One of the noted authors is a group or consortium [program collaborators for Environmental influences on Child Health Outcomes]. In addition to naming the author group, please list the individual authors and affiliations within this group in the acknowledgments section of your manuscript. Please also indicate clearly a lead author for this group along with a contact email address.

We referred to the authors for ECHO in the Acknowledgment section. Institutions were also listed. There is not a lead author for this group.

The ethics statement only appears in the revised Methods section.

7. Please upload a copy of Supporting Information (Supplementary Table 1,2,3,4) which you refer to in your text on page 11 and 12.

The Supplementary Tables are uploaded.

We confirm the accuracy of the references.

Reviewer #1: The manuscript presents an attempt to evaluate the relationships of SARS-CoV-2 serology, self-reported infection, and pandemic-related stress with birth outcomes in largely urban cohorts from three ECHO cohort centers.

The pregnancy outcomes are well-defined and include gestational age (GA), birth weight (BW), and BW Z-score for gestational age (as continuous variables. Categorical outcomes included prematurity,

low birth weight (LBW) and small for gestational age (SGA).

The analyses did not document any differences in gestational age, birth weight, PTB, or LBW. Nearly identical results were obtained when comparing birth outcomes by serology status. The only positive association was the finding that children with lower birth weight and BW Z-score for GA (were more likely to deliver preterm in univariate models.

The study is performed in an elegant way and I do not have any major questions regarding the methodology.

Thank you for the kind words!

The only point worth explaining is the way the final sample was identified. It is said that the study was nested within a subsample of mothers and newborns enrolled in three participating ECHO cohorts. Why three different cohorts were utilized? What were the criteria for inclusion and exclusion?

This was a convenience sample of three cohorts which chose to participate as a supplementary funding opportunity. We have clarified this in the manuscript.

Reviewer #2: Here the authors aim at understanding the effect that SARS-CoV-2 infection has during or before pregnancy. In an observational study including three urban cohorts that gave birth between January 2020 to September 2021 they analyzed samples from 211 mothers for the presence of IgG1, IgG2, IgG3, IgG4, IgM and IgA subclass antibodies against N, S1, S2 and RBD areas of the spike protein. Pre-pandemic samples (n=234) were used to define non-infected antibody levels, and levels between 3-6 SD above background were classed as indeterminate whereas over 6 SD were considered positive. All mother with at least one subclass over 6SD were considered positive, mothers with a least on subclass in the 3-6 SD as indeterminate, and all classes below 3SD as negative. Seropositivity was seen in 36% of mothers, while 43% reported that they had been infected. IgG1 total seropositivity as well as reactivity against N and RBD was modestly correlated with reported infection. Neither serology + self reported infection nor serology alone was correlated with gestational age, birth weight, pre-term birth or low birth weight. IgM positivity was scored in 9 mothers, and in this case a significant decrease in birth weight was observed. This could not be solely explained by a posttraumatic stress score calculated from the mothers. In addition, in children born from mothers before any vaccines was available, decreased birth weights was also seen children born to IgM positive mothers.

During COVID-19 infection it has been shown that IgM antibodies occur in serum at approximately the same time as IgG and IgA antibodies. Since IgM then decrease more rapidly the data indicate that recent (possibly during pregnancy) infections with COVID-19 can influence pregnancy outcomes, but the authors argue caution as this result is only based on very few cases (n=9).

Overall, the study is of interest although its main conclusion is based on very few samples. It is well written, the data is presented in a clear and careful way, and the results are discussed cautiously. Although the data will not overall change the general recommendation to try to avoid infection with SARS-CoV- 2 during pregnancy, it gives some further support to the recommendation.

Specific points

I think the material is big enough to not only discuss the potential correlation between IgM antibodies and pregnancy/birth related problems but also the lack of any correlation to IgG levels. This could suggest that it is unlikely that pre-pregnancy (or even early pregnancy) infections will not interfere with outcomes. I fully understand that it is hard to rule out that this could be due to early problems are not scored (i.e. problems conceiving or early spontaneous abortion), but at least late problems from previous infection are unlikely. I would suggest a short paragraph about this in the discussion.

We have added the requested clarification.

On line 234, the sentence “When we compared mothers by immunoglobulin subtype (Table 2c), the small number of IgM seropositive mothers in the sample (n=9) had children with 407 grams lower birth weight (95% CI: 85-728) and BW Z-score for GA (0.65 SD; 95% CI 0.002-1.30) were more likely to deliver preterm (OR 6.29, 95% CI 1.12-25.2) in univariate models.” Is very hard to understand – please reformulate.

We have clarified as requested.

---

## [Editor Report · Decision Letter 1]

18 Oct 2023

Associations of SARS-CoV-2 Antibodies with Birth Outcomes: Results from Three Urban Birth Cohorts in the NIH Environmental Influences on Child Health Outcomes Program

PONE-D-23-20437R1

Dear Dr. Trasande

We’re pleased to inform you that your manuscript has been judged scientifically suitable for publication and will be formally accepted for publication once it meets all outstanding technical requirements.

Kind regards,

Faten Abdelaal Okda

Academic Editor

PLOS ONE
---

## [Editor Report · Acceptance letter]

13 Nov 2023

PONE-D-23-20437R1 

Associations of SARS-CoV-2 Antibodies with Birth Outcomes: Results from Three Urban Birth Cohorts in the NIH Environmental Influences on Child Health Outcomes Program 

Dear Dr. Trasande:

I'm pleased to inform you that your manuscript has been deemed suitable for publication in PLOS ONE. Congratulations! Your manuscript is now with our production department. 

Kind regards, 

on behalf of

Dr. Faten Abdelaal Okda 

Academic Editor

PLOS ONE